# High-resolution digital elevation models and orthomosaics generated from historical aerial photographs (since the 1960s) of the Bale Mountains in Ethiopia

Mohammed Ahmed Muhammed[1,2], Binyam Tesfaw Hailu[2,3], Georg Miehe[4], Luise Wraase[1], Thomas Nauss[1], Dirk Zeuss[1]

[1]Department of Environmental Informatics, Faculty of Geography, Philipps-Universität Marburg, Deutschhausstraße 12, 35032 Marburg, Germany
[2]Remote Sensing and Geo-Informatics Stream, School of Earth Sciences, College of Natural and Computational Science, Addis Ababa University, Addis Ababa, 1176, Ethiopia,
[3]Department of Geosciences and Geography, University of Helsinki, PO Box 64 (Gustaf Hällströmin katu 2), FI-00014 Helsinki, Finland
[4]Department of Geography, Vegetation Geography, Philipps-Universität Marburg, Deutschhausstraße 10, 35032 Marburg, Germany

*Correspondence to*: Mohammed Ahmed (mohammed.ahmedgis@aau.edu.et, mohammed.muhammed@geo.uni-marburg.de)

**Abstract.** The natural resources of Ethiopian high-altitude ecosystems are commonly perceived as increasingly threatened by devastating land-use practices owing to decreasing lowland resources. Quantified time-series data of the course of land-use cover changes are still needed. Very high-resolution digital data on the historical landscape over the recent decades are needed for determining the impacts of changes in afro-alpine ecosystems. However, digital elevation models (DEMs) and orthomosaics do not exist for most afro-alpine ecosystems of Africa. We processed the only available and oldest historical aerial photographs for Ethiopia and of any afro-alpine ecosystem. Here, we provide each a DEM and orthomosaic image for the years 1967 and 1984 for the Bale Mountains in Ethiopia, which comprise the largest afro-alpine ecosystem in Africa. We used 298 historical aerial photographs captured in 1967 and 1984 for generating DEMs and orthomosaics with a Structure from Motion Multi View Stereo Photogrammetry workflow along an elevation gradient from 977 to 4,377 m above sea level (asl) at a very high spatial resolution of 0.84 m and 0.98 m for the years 1967 and 1984, respectively. The Structure from Motion Multi View Stereo Photogrammetry workflow, employed with Agisoft Metashape, represents a modern approach that combines computer vision and photogrammetry. This method proves useful for reconstructing DEMs and Orthomosaics from historical aerial photographs, with a focus on high spatial resolution. To validate the accuracy of the reconstructed DEMs, ground control points gathered through GPS measurements were used, resulting in RMSE values of 3.55 m for the year 1967 and 3.44 m for the year 1984. Our datasets can be used by researchers and policymakers for watershed management, as the area provides water for more than 30 million people, landscape management, detailed mapping and analysis of geological and archaeological features as well as natural resources, analyses of geomorphological processes, and biodiversity research.

# 1 Introduction

Landscapes worldwide are increasingly affected by anthropogenic influences and natural factors such as land-use and climate change (Bendix et al., 2021; He et al., 2021; Huggel et al., 2012; Peters et al., 2019; Slaymaker and Embleton-Hamann, 2018; Turner and Gardner, 2015). Afro-alpine ecosystems are particularly threatened due to population growth and expansion of human settlements, overgrazing, recurrent fire, deforestation, agricultural expansion (Gehrke and Linder, 2014; Gil-Romera et al., 2019; Kidane et al., 2012; Mezgebu and Workineh, 2017; Muhammed and Elias, 2021; Nyssen et al., 2014), and climate change (Colwell et al., 2008; Diaz and Bradley, 1997; Jacob et al., 2020; Kidane et al., 2022; Palomo, 2017). These impacts are of particular importance in afro-alpine ecosystems because they are hotspots of biodiversity and endemism (Gehrke and Linder, 2014; Merckx et al., 2015), and thus prominent for their ecological significance and considerable economic, recreational, aesthetic, and scientific value (Muhammed and Elias, 2021; Rahbek et al., 2019). They also constitute important freshwater sources for mountain and lowland ecosystems, as well as for millions of people living in the adjacent areas. In the year 2015, > 30% of the global population lived in mountain areas (Thornton et al., 2022).

For analysing the effects of land-use and climate change in afro-alpine ecosystems, digital data on the historical landscape at very high spatial resolution is needed. Detailed digital elevation models (DEMs) and orthomosaics are important data sources for quantifying the impacts of anthropogenic and natural changes. For instance, DEMs and orthomosaics can be used for research in structural geology (Da Costa and Starkey, 2001), earthquake impacts (Lu et al., 2021), archaeology (Risbøl et al., 2015), geomorphology (van Westen and Lulie Getahun, 2003), water resources management (Chignell et al., 2019), and land use land cover dynamics (Jacob et al., 2016). However, most studies have used satellite images with 10–60 m resolution (Kidane et al., 2012; Mezgebu and Workineh, 2017; Muhammed and Elias, 2021), freely available DEMs of 30 m or coarser resolution (Chignell et al., 2019; Farr et al., 2007; Friss et al., 2010; Kidane et al., 2022), and georeferenced aerial photographs with small spatial extent (< 225 km$^2$; Carta et al., 2018; Jacob et al., 2016; Johansson et al., 2019). However, no openly accessible historical DEMs and orthomosaics at high spatial resolution (< 5 m) and large spatial extent (> 127 km$^2$) exist for any afro-alpine ecosystems. Currently, DEMs and orthomosaics generated with unmanned aerial vehicles or helicopters reach spatial resolutions of 0.1 to 0.5 m with an extent ranging from 1.6 to 127 km$^2$ only in urban regions (Benoit et al., 2019; Bühler et al., 2012; Immerzeel et al., 2014).

Globally, very high-resolution satellite programmes started from 1960, e.g. the CORONA satellites, which were the first spy series of very high spatial resolution (1.8 – 7.5 m) to capture stereo photographs for military purposes from 1960 – 1972 (Altmaier and Kany, 2002; Day et al., 1998). These images were declassified for civilian use in 1995 (Dashora et al. 2007). However, invisibility of film reference data, low texture and film saturation, inconsistent overlap, and scanning artefacts are their limitations (Ghuffar et al., 2022). Later, commercial satellites emerged, e.g. the Indian Remote Sensing Satellites (IRS; since 1995) at 5 m, IKONOS (since 1999) at 0.8 m, QuickBird (since 2001) at 0.65 m, Worldview 1- 4 (since 2007) at 0.3 m, Pléiades 1A and 1B (since 2007) at 0.5 m and Satellite Pour l'Observation de la Terre (SPOT 6/7; since 2012) at 1.5m spatial resolution. Our scanned aerial photographs of the years 1967 and 1984 spatially align and could be used as a snapshot of data continuity for the global coverage (Turner et al., 2015) between the times of Corona satellites in the 1960 and the start of the IRS and IKONOS satellite in the early 1990s.

To reconstruct detailed DEMs and Orthophotos, traditional aerial triangulations were employed. This involved determining the optimal initial values for camera poses and 3D points, alongside the utilisation of vertically captured aerial photographs, a practice that remained prevalent until the mid-nineties. The traditional aerial triangulation method consumes more time during the pre-processing stage (Schenk, 1997). Then, automatic aerial triangulation was created. However, the well-established technique and algorithm known as Structure from Motion (SfM) in conjunction with a multi-view stereo photogrammetry (MVS) workflow simplifies the process. It allows for the automatic and simultaneous recovery of all unknown parameters related to camera poses and 3D points from overlapping images (Snavely et al., 2008), whether they are captured with digital or non-digital cameras. SfM-MVS Photogrammetry is a low-cost and efficient method that selects high-quality images, offering an accurate and effective approach. This recently emerged algorithm is user-friendly and particularly well-suited for generating high-resolution topographic reconstructions, especially when working with scanned historical aerial photographs (AgiSoft LLC, 2021; Nyssen et al., 2022; Sevara et al., 2018; Westoby et al., 2012).

In this study, we aim to present DEMs and orthomosaics for the years 1967 and 1984 at very high spatial resolutions (0.84 m and 0.98 m, respectively) for the remote Bale Mountains in Ethiopia, which comprise the largest afro-alpine ecosystem in Africa (Chignell et al., 2019). The quality of both the 1967 and 1984 DEMs were assessed based on ground control points (GCPs) collected from the field. In addition, comparison between the generated DEMs and the five readily available global DEMs were done. Our data can be used by researchers and policy makers for watershed management, analyses of historical landscape change, detailed mapping and analyses of geological and archaeological features, as well as natural resources, (4) analyses of geomorphological processes, socioecological patterns and dynamics, modelling and planning for telecommunications, and biodiversity research. Owing to its high spatial resolution and spatiotemporal coverage, our data will foster studies on the drivers of landscape change and its quantification.

## 2 Material and Methods

### 2.1 Study area

The study area lies between 6º20′27″ to 7º51′17″ N and 39º16′58″ to 40º04′40″ E, (approximately 5,730 km²) located 400 km southeast of Addis Ababa in the Oromia regional state of Ethiopia (Fig. 1). The area lies within the Bale Mountains and covers 25 % of the afro-alpine ecosystems of Africa (de Deus Vidal and Clark, 2020; Carbutt, 2020; Gehrke and Linder, 2014). The Bale Mountains were formed by lava outpourings of the Trappean series, which created a vast lava plateau considerably eroded and flattened by later glaciations (Berhe et al., 1987; Groos et al., 2021; Williams, 2017). The soils in the area tend to be shallow, gravelly, and fertile silty loams of reddish-brown to black colour, as the parent rock is mainly basaltic and trachytic (Miehe and Miehe, 1994). The study area provides the water source for over 30 million people in Ethiopia, Somalia, and Kenya, and contributes to five major perennial rivers (Fig. 1).

The climate in the study area varies from north to south mainly owing to differences in elevation, aspect, and the influence of lowland hot air masses, which also influence the annual migration of the Intertropical Convergence Zone and Indian Ocean Monsoon (Miehe and Miehe, 1994). The mean monthly minimum and maximum temperatures are 5.6 °C and 21.4 °C, respectively, for an altitudinal range of 2,700 to 4,377 m above sea level (asl).

The mean annual ground temperature ranges from 7 to 11 °C (Groos et al., 2022). The lowest and highest temperatures recorded on the Sanetti Plateau were -15 ºC and 26 ºC, respectively (Hillman, 1988), where frosts occur during all clear nights throughout the year (Groos et al., 2022). Rainfall is variable throughout the study area and ranges from 800 to 1,500 mm annually (Woldu et al., 1989).

The study area is known for plenty of vascular plant species ranging from tropical rainforests to afro-alpine ecosystem. It is classified in to five vegetation zones; 1) Gaysay valley grasslands: covered by small grassland situated within a broad flat valley at an altitude of 2,600-3,000 m asl in the northern part of the study area; 2) Dry evergreen montane forest: ranging from 2,600 and 3,200 m asl dominated by *Juniperus procera* and *Hagenia abyssinica*, 3) Afro-alpine: reside in the central part of the study area including Sanneti Plateau between about 3,200 and 4,377 m asl characterised by short, herbaceous plant communities dominated by *Alchemilla*, *Helichrysum* and *Artemisia* shrubs; 4) *Ericaceous* belt: spans most of the escarpment areas between 3,100-3,800 m asl and forms belt between the high altitude Afro-alpine plateau and the lower Harenna forest. The vegetation in this belt is dominated by *Erica arborea* and *Erica trimera* species scattered up to 4,276 m asl, 5) Harenna moist montane forest: this forest is one of the most extensive and largely natural forests remaining in Ethiopia. It covers the southern slope of the mountains, extending between 1,450 to 3,200 m asl (Asefa et al., 2020; Kidane et al., 2019; Miehe and Miehe, 1994; Fig. 1). Topographically, the study area has a maximum slope of 58.63°. Almost 50% of the area is exposed to east, south-east and south facing slopes.

**2.2 Data**

We used and processed historical aerial photographs (HAPs) taken in 1967 and 1984. They have a forward overlap of approximately 60 %, a side overlap of approximately 30%, and were captured using an aerial 9 x 9 inch film frame camera, which was scanned at a resolution of 1,200 dots per inch. The qualities of all scanned HAPs were excellent enough for processing with an image quality value above 0.5 units. Worldview-1 satellite imagery (DigitalGlobe Foundation) was used for the extraction of on-image selected GCPs and applied for pre-processing (Groos et al., 2021). A subset of the GCPs was used for internal accuracy assessment. For assessing the external accuracy of our generated DEMs and orthomosaics, we used ~~ground control points~~ GCPs collected in the field and additionally from five sources of data: (1) Advanced Spaceborne Thermal Emission and Reflection Radiometer Global Digital Elevation Model (ASTER GDEM) data with a spatial resolution of 15 m, (2) data from the Shuttle Radar Topography Mission (SRTM) with a spatial resolution of 30 m, ~~and~~ (3) Advanced Land Observing Satellite Phased Array Type L band Synthetic Aperture Radar data (ALOS PALSAR) with a spatial resolution of 12.5 m, (4) TanDEM-X with a spatial resolution of 90 m from (Rizzoli et al., 2017), and (5) Copernicus DEM (COP-DEM) with a spatial resolution of 30 m (European Space Agency and Airbus, 2022). Each of the five downloaded DEMs was projected on the Adindan UTM coordinate system zone 37 N (EPSG: 20137) using ArcGIS Desktop (Version 10.8.2) (ESRI Inc., 2021).

The 1967 HAPs were acquired by the United States Air Force under Project number AF 58-3, using a four-engine RC-130A aircraft flying at an altitude of 31,000 feet asl equipped with a KC-1(B) PLANIGON camera. For camera and lens number, calibrated focal length, magazine serial number, roll number, exact acquisition dates, and flight index map (see supplementary material Fig. S1 and Table S1). The shutter speed and aperture were 1/150 s and F-8, respectively, resulting in a scale of approximately 1:62,000.

The 1984 HAPs were obtained by SWEDSURVEY Company under project number "ET 1:5" using a Wild RC 10 camera (Wild Universal Aviogon II lens type, camera serial number 3045) with a calibrated focal length of 152.822 mm and a maximum aperture of F/4.0. The flight campaign had an average altitude of 7,600 m asl and was conducted from 15–17 January, 1984. For more details on roll number, strip number, photo number, acquisition date, and flight index map see supplementary material Fig. S2 and Table S2.

### 2.3 Data preparation

The data preparation step included extraction of the calibrated focal length, camera principal point coordinates, camera exposure stations, fiducial marks, and ground control points. All data preparation steps were performed with Agisoft Metashape Professional (Version 1.8.0.13794) (AgiSoft LLC, 2021), Google Earth, and ArcGIS Desktop (Version 10.8.2) (ESRI Inc., 2021).

### 2.3.1 Calibrated focal length

The calibrated focal length is a numerical best-balanced value used to determine the scale of the photograph. It is computed from the equivalent focal length to obtain minimum distortion and to match the lens distortion (Clarke and Fryer, 1998). However, the Structure from Motion with Multi View Stereo workflow does not require the principal distance of the camera/lens at the time of exposure for reconstruction workflow, inputting the calibrated focal length of the different blocks of HAPs will solve the geometry and calculate the camera positions and create 3D coordinate points. Alternatively, the method employs a default focal length value of 35mm (AgiSoft LLC, 2021). The calibrated focal lengths were extracted from meta information on the HAPs of the film and from Spriggs (1966).

### 2.3.2 Principal points

A principal point is a point in the precise centre of a photograph. Its coordinates indicate the perpendicular interception of the optical axis of the lens with the sensor plane (supplementary material Tables S5 and S6). For the year 1967, easting and northing of the principal points (camera positions) were digitized from the scanned and georeferenced topographic flight index map, which included points of camera positions and polygons of consecutive aerial photograph coverage. Elevation data of the principal points were extracted using Google Earth. For the 1984 HAPs, the principal points were extracted from aero-triangulation documents in the Geospatial Information Institute of Ethiopia. However, the camera pose is estimated in a Structure from Motion Multi View Stereo workflow and not necessarily needed. Both the film principal points and camera positions play a vital role in the workflow by enhancing the quality of the image alignment step to the reconstruction process. For more details, see supplementary material Tables S3 and S4.

### 2.3.3 Fiducial marks

Fiducial marks are markers rigidly connected to the centre or corners of the camera body. When the film is exposed, these marks appear on the film negative. Fiducial marks and their coordinates were used as input for the processing of scanned HAPs during the camera calibration workflow stage. Fiducial marks present in the scanned images were utilised to rectify film shrinkage and linear deformation. This incorporation enhanced the

robustness of the reconstruction process. They were used for the calculation of internal camera calibration parameters. For more details, see supplementary material Figs. S3 and S4 along with Tables S5 and S6.

### 2.3.4 Ground control points

Ground Control Points (GCPs) are locations on the ground localized by a spatial coordinate reference system. GCPs have dual purpose as they can be used to reconstruct the digital terrain model and/or to determine the accuracy and impact of processing parameters. Without the use of GCPs the reconstruction from the imagery could have errors in scale and orientation, as well as incorrect absolute position information. Thus, they were used to georeference the aerial camera data. 76 on-image selected GCPs were extracted from Worldview-1 sat-

ellite imagery (DigitalGlobe Foundation) from (Groos et al., 2021) and Google Earth. These included 49 and 27 GCPs used to georeference the 1967 and the 1984 HAPs, respectively, as their acquisition and flight index differed. For more details, see supplementary material Table S7.

## 3 Data processing

Data processing was performed using Structure-from-Motion Multi View Stereo Photogrammetry methodology

by assigning appropriate values and settings in the main steps of loading the aerial photographs, aligning the cameras, building a dense point cloud, a mesh, a digital elevation model, an orthomosaic, and lastly exporting all the results (Fig. 14).

### 3.1 Structure-from-Motion Multi View Stereo Photogrammetry

Traditional digital photogrammetry requires the 3D position and camera poses, or the 3D position of a series of

205 control points to be known to reconstruct 3D structure from 2D overlapping photographs. Structure-from-Motion (SfM) is the process of estimating the 3D structure of a scene from a series of 2D images with the absence or presence of camera exposure location and GCPs. SfM uses the Scale Invariant Feature Transform algorithm (Lowe, 2004) and Hessian-based detectors (Lindeberg, 1998) to address the problem of identification of features in individual images used for image correspondence especially for the reconstruction of DEMs and

210 orthomosaics using scanned historical aerial photographs. In this step camera pose and scene geometry are reconstructed in the automatic identification of matching features in multiple images. In our workflow SfM is a specific step that provides camera parameters and a sparse point cloud (Daniotti et al., 2020). Then, dense image matching algorithms, such as Multi View Stereo (MVS), are used in a subsequent step to densify the point cloud. MVS will capture more angles in between instead of taking two photos from two different viewpoints to

215 increase durability, for example for image noise or surface texture (Wu et al., 2018). The MVS algorithm handles images with more different perspectives, such as sets of images that surround objects, and can also handle very large numbers of images (Fig. 13).

The whole above mentioned process is referred to as SfM MVS Photogrammetry; a workflow methodology including multiple different algorithms, computer vision, photogrammetry and more conventional survey tech-

220 niques (Eltner and Sofia, 2020). It is applicable for soil erosion, volcanology, glaciology, coastal morphology, mass movements, and fluvial morphology. It was used in at least 65 scientific studies from 2012 to 2015 (Eltner et al., 2016) and also recently (Grottoli et al., 2020; Nyssen et al., 2022; Tomczyk and Ewertowski, 2021). The

advantages of SfM MVS Photogrammetry are that it can be applied fully automated (Eltner et al., 2016), is flexible, efficient, inexpensive, and requires little training (Iglhaut et al., 2019; Westoby et al., 2012). The following steps were sequentially applied for generating DEMs and orthomosaics with SfM MVS Photogrammetry. The whole process is also shown in Fig. 14.

**Aligning**. This was the first step in the model generation process after data acquisition to generate tie points and match corresponding features using the Scale Invariant Feature Transform algorithm (Lowe, 1999). It identifies and extracts the important points present in the photographs by interest operators using different characteristics like lighting, colour, and rotation. Once identified, the exact same points are coupled using the concept of Euclidean distance.

**Bundle adjustment.** Bundle adjustment was used for calculating individual camera positions and relative positions of corresponding features (Triggs et al., 2000). It is an approach in the SfM workflow that simultaneously recovers 3D structure, camera pose, and even the intrinsic camera parameters. Thus, bundles of rays connecting the camera centres to 3D scene points and an adjustment in terms of iteratively minimizes the reprojection error. A least-squares approach was used for estimating camera poses and 3D points presented by Eq. (1).

$$\min_{\boldsymbol{a}_j, \boldsymbol{b}_i} \sum_{i=1}^{m} \sum_{j=1}^{n} d\left(\boldsymbol{Q}(\boldsymbol{a}_j, \boldsymbol{b}_i),\ \boldsymbol{X}_{ij}\right)^2 \tag{1}$$

Assuming that $n$ 3D points are seen $m$ views and let $X_{ij}$ be the projection of the $i^{th}$ point on image $j$. Where $\boldsymbol{Q}(\boldsymbol{a}_j, \boldsymbol{b}_i)$ is the predicted projection of point i on image j and d (**x, y**) denotes the Euclidean distance between the image points represented by the inhomogeneous vectors **x** and **y**.

**Multi view stereo matching.** Dense point clouds (multi view stereo) and 3D surfaces (build mesh) were calculated using known camera parameters (camera focal length, coordinates of the image principal point, lens distortion coefficients, omega, phi, and kappa) and with the SfM points as ground control. The Agisoft Metashsape professional software uses a combination of area based matching and feature based matching algorithms to reconstruct the dense image. All pixels in all images were used, so the resulting dense model was similar in resolution to the raw photographs. The dense point clouds were classified using parameters of maximum angle, maximum distance and cell size for automatic division of all points.

**Georectification.** In this stage, the point cloud from an internal, arbitrary coordinate system was projected onto a geographical coordinate system. The 3D point clouds that were generated through SfM-MVS photogrammetry are in a relative image-space coordinate system and need to be aligned to a real-world object-space coordinate system. This transformation can be achieved using a 3D similarity transform based on either known camera poses, a small number of GCPs with known object-space coordinates, or a combination of the two. For the 1984 HAPs, the georectification process was completed using the camera positions and focal lengths, and for the 1967 HAPs by incorporating on-image selected GCPs with known coordinates.

**Derivative product generation.** In this step, the final DEM and orthomosaics were created from the dense point clouds and 3D surfaces. The DEMs were generated from classified ground dense point clouds used as input whereas the orthomosaics were generated using a mesh as input. For more details on the inputs and values used, see supplementary material Table S9.

## 4 Results and discussion

The produced tie points resulted from the "align" step consisted of 412,903 (1967) and 450,071 (1984) filtered points, with low tie point reprojection errors ranging from 1.49 to 0.57 pixels, demonstrating the high quality of the image geometry network (Table 1). The dense point clouds consisted of 4,821,064,653 points for the 1967 HAPs and 6,501,301,603 points for the 1984 HAPs. These points were classified into ground points using the classification parameters of maximum angle, maximum distance and cell size of 37.5°, 0.5 m and 30 m respectively. This approach takes the ruggedness of the topography and different coverage of vegetation of the study area into consideration and improves the accuracy. The classified ground dense point clouds and the mesh were used to obtain a DEM and orthomosaic, respectively, with spatial resolutions of 0.84 m and 0.98 m for 1967 and 1984 data (Figs. 2, 3 and 12). The resulting very high resolution DEMs showed the terrain and ruggedness clearly compared to other freely available low resolution DEMs used for comparison and assessment (Fig. 12).

In a previous study, Frankl et al. (2015) processed 27 HAPs (both vertical and oblique) acquired in 1935 in the Suluh River Valley, Ethiopia, using SfM MVS and produced orthomosaics with planimetric and elevation root mean square error (RMSE) accuracy of 30 and 50.7 m, respectively. However, the accuracy of our DEMs produced for the years 1967 and 1984 was higher, with an RMSE of 3.55 and 3.44 m, respectively (Table 2). This is due to the effect of different types of vegetation cover and terrain morphology (Aguilar et al., 2005; Spaete et al., 2011). Considering that the SfM–MVS method yields accuracies of 1/1000 of the viewing distance, the elevations above the ground surface at which our HAPs were collected (8,300 m for 1967 and 5,068 m for 1984) yield acceptable error values of 8.3 and 5 m, respectively, which were achieved with our data products.

The orthomosaics and DEMs provided here will help to identify areas where changes in environmental characteristics occurred (e.g., encroachment of humans or deforestation of the Ericaceous vegetation in the north western part of the study area). For example, it became clear from our data that there was no severe human interference and deforestation before 1967. However, the 1984 orthomosaics provide evidence that people began intruding into Bale Mountains National Park and degraded the land by clearing vegetation near the settlements (Figs. 7 and 8).

### 4.1 Use Case studies

The generated datasets can be used for different applications and we give some brief examples of five use cases here. Firstly, within our study area, in two locations soil material was excavated and used for constructing and repairing a gravel road (Fig. 4). An example volumetric analysis was done with Agisoft Metashape Professional Edition software (AgiSoft LLC, 2021) for the selected sites. The volumetric change calculated from the 1967 and 1984 DEMs revealed the removed soil was 14,728.9 m³, for data example 1 (Fig. 4a, b) and 97,352 m³ for data example 2 (Fig. 4e, f).

Secondly, the number of settlements in the afro-alpine ecosystem and central parts of the Bale Mountains National Park increased (Fig. 7). There was not much human interference and deforestation until 1967. However, encroachment of humans and degradation of the vegetation was visible starting from 1984 onwards (Figs. 7 and 8). This is complemented with Reber et al. (2018), who found 870 settlements and 331 rock shelters only in the afro-alpine ecosystem of the Bale Mountains National Park.

Thirdly, the resulted orthomosaic datasets could be used as an input to assess the historical distribution of *Ericaceous* species to today's recent distribution in the Bale Mountains, and in particular to detect the historical de-

velopment of the *Erica* patches scattered across the Sanneti Plateau given the detailed resolution of our datasets.
In addition, Mekonnen et al. (2023), examined potential factors responsible for the now fragmented occurrence of *Erica* at the Sanetti Plateau,, which is possibly also attributed to the topography of the area (Mekonnen et al., 2023). The distribution of the giant root-rat over the Sanneti Plateau and adjacent valleys in the north-western parts of the study area was predicted by (Wraase et al., 2023). However, this study could be extended in the future by analysing the historical distribution shift of the giant root-rat and further explaining its effect on the landscape and its ability for landscape change.

Fourthly, the very high-resolution information from the 1984 orthomosaic can be employed for river and river side altimeter change assessments (Fig. 9). Already visible by using the orthomosaics is that the river named Welmel Tika and its side changed both in width and altitude, thereby reducing previously forested area.

Lastly, our generated datasets are useful for studying changes in surface geomorphology and urban areas. Currently, at the north western corner of the study area above Adaba town, there is an artificial lake dam called Melka Wakena built for hydroelectric power generation. Back in 1967 the area was cultivated land and had a river course inside today's dam (Fig. 10). The urban area of Adaba town in the north-western part of the study area, e.g., also expanded from 53.83 ha in 1967 to 107.09 ha in 1984 and 475.79 ha in 2011 (Fig. 11).

### 4.2 Quality assessment

The quality of our generated DEM dataset was assessed based on external and internal accuracy assessment. For *external* accuracy assessment, the quality of the resulting DEM dataset was assessed by using external sources (509 GPS control points [CPs] collected in the field as reference points, as well as ASTER GDEM (30 m), SRTM DEM (30 m), COP-DEM (30 m), TanDEM-X DEM (90 m) and ALOS PALSAR (12.5 m) points for both 1967 and 1984. The values of elevation (in m) for each CP from each of the DEMs were extracted, and then accuracy assessment was done using linear regression models by comparing the descriptive statistics of elevation, minimum elevation difference with field control points, first quantile, median, third quantile, maximum elevation difference from CPs, residual standard error, multiple R-squared correlation coefficient, and root mean square error. Comparing the 1967 and 1984 DEMs with external globally freely available DEMs is beneficial and gives additional information regarding the accuracy of DEMs with complex topography like present in our study area.

Our generated DEMs have higher accuracies than previous DEMs (Fig. 5 and Table 2). As demonstrated by its RMSE value of 3.44, the DEM generated from the 1984 HAPs had the best quality. The 1967 DEM also had a very good quality, as indicated by an RMSE value of 3.55. The multiple R-squared value was 0.9998 (equal to the 1984 DEM) and there was no significant difference between 1967 and 1984 DEMs (Table 2). Our datasets are accurate for doing an in-depth analysis as the spatial resolution is very high (0.84 and 0.98 m for the 1967 and 1984, respectively) compared to hitherto freely available DEMs and satellite images. In addition, our datasets are spatially aligned to other global images and relate to external reference systems too (Fig. 6).

In addition, the resulting DEM dataset was also assessed by using on-image selected GCPs for *internal* accuracy assessment (12 out of 49 GCPs and 7 out of 27 GCPs for the 1967 and 1984 HAPs, respectively; Table 1). It indicates a high quality of internal image network geometry as huge dense point clouds for historical aerial photographs. This was illustrated by the low sub-pixel values of tie point reprojection errors of 1.49 and 0.68 pixels and the RMSE of 4.24 and 0.88 cm on the control points for the years 1967 and 1984, respectively.

Among the five readily available DEMs used for comparison, Copernicus-DEM (30 m) had the highest accuracy (RMSE value of 4.9) complement with the validation maximum RMSE value of 5 (European Space Agency and Airbus, 2022). In similar high mountain areas, carried out to evaluate the performance of DEMs for tectonogeomorphic analysis in the South American Andes, the Copernicus DEM (30 m) also performed best (Del Rosario González-Moradas et al., 2023). Thus, complementing the comparison made for freely available DEMs for our study area, it showed the best accuracy (Fig. 5 and Table 2). However, SRTM DEM had the next most accurate (RMSE value of 5.64) in our study area, which has a complex topography and a huge elevation difference of 3800 m. These results agree with Sena et al. (2020). The SRTM DEM showed better precision (Fig. 5 and Table 2). However, others (Shebl and Csámer, 2021; Chowdhuri et al., 2021; Jalal et al., 2020) concluded that ALOS PALSAR is more accurate than the SRTM and ASTER DEMs and obtained different results for our study area. The ALOS PALSAR DEM (Fig. 5 and Table 2) was the third most accurate DEM with an RMSE vertical accuracy of 11.38. The ASTER GDEM was the fourth accurate in the area as the residual standard error value was 11.42 with an RMSE value of 11.54, indicating that there is a ~~huge~~ gap in elevation values compared to the field GPS control points (Fig. 5). Thomas et al. (2014) also attested to a higher accuracy for SRTM compared to ASTER GDEM. The TanDEM-X 90m (Rizzoli et al., 2017) DEMs was the least accurate with an RMSE value of 11.9. However, the TanDEM-X 90 is more accurate compared to Multi-Error-Removed Improved-Terrain DEM and SRTM 90 m resolution accuracy assessment checked within six continents at 32 floodplain locations using high resolution LiDAR DEMs as a reference(Hawker et al., 2019).

**5 Data availability**

All described datasets are available in the Zenodo link repository https://doi.org/10.5281/zenodo.7271617 (Muhammed et al., 2022a) for the inputs and https://doi.org/10.5281/zenodo.7269999 (Muhammed et al., 2022b) for the results obtained.

The structure of the dataset is as follows.

1   Unprocessed scanned aerial Photographs (approximately 36.1 GB) are available at https://doi.org/10.5281/zenodo.7271617 (Muhammed et al., 2022a). The images are zipped into four zipped folders named: "1967_Scanned_HAPs_Part1.7z" and "1967_Scanned_HAPs_Part2.7z" for the 1967 historical aerial photographs; and "1984_Scanned_HAPs_Part1.7z" and "1984_Scanned_HAPs_Part2.7z" for the 1984 historical aerial photographs. The scanned photographs are in Tiff format except four photographs in JPEG format.

2   Camera position coordinates (principal point coordinates) data are zipped into one folder "Camera_Position.zip" and two text files: "PP_1967.txt" and "PP_1984.txt" available at https://doi.org/10.5281/zenodo.7271617 (Muhammed et al., 2022a).

3   Flight index shapefile data are zipped into one folder named "Flight_Index.zip" with four shape files: "PP_1967.shp", "PP_1984.shp", "flight_index_1967.shp", and "flight_index_1984.shp"; and ground control point data are zipped into one folder named "GCP.zip" with two text files: "GCP_1967.txt" and "GCP_1984.txt available at https://doi.org/10.5281/zenodo.7271617 (Muhammed et al., 2022a).

4   The results of photogrammetric processing (approximately 32.5 GB) are available at https://doi.org/10.5281/zenodo.7269999 (Muhammed et al., 2022b), and are grouped into subfolders named: "DEM_1967.7z": inside the zipped folder "1967_DEM.tif" (digital elevation model for the year

1967), "DEM_1984.7z": inside the zipped folder "1984_DEM.tif" (digital elevation model for the year 1984), and "1967_Orthomosaic.7z" and "1984_Orthomosaic.7z" contain orthomosaic files "1967_orthomosaic.tif" and "1984_orthomosaic.tif" for the year 1967 and 1984, respectively. All data are in GeoTIFF format in the Adindan UTM Zone 37 N (EPSG: 20137) projected coordinate system.

5    Data extracted from the resulted study area for the volumetric calculation and visualization (approximately 1 MB) are available at https://doi.org/10.5281/zenodo.7269999 (Muhammed et al., 2022b) inside zipped folder "Data_Examples.Zip" and "Accuracy_assessment.7z" contains text and excel file used for accuracy assessment calculation. For more detail on the list of filenames for corresponding years and content descriptions, see supplementary material Table S8.

## 6 Conclusions

In our study area, we analyzed the sole available and earliest historical aerial photographs for Ethiopia, which, to the best of our knowledge, encompass not only our specific study area but also the entire afro-alpine ecosystem. We decided to use Agisoft Metashape/SfM-MVS Photogrammetry because the method is based on a newly developed algorithm and photogrammetry. We generated the first historical very high-resolution DEMs and orthomosaics for the years 1967 and 1984 at a larger spatial extent (5,730 km$^2$) and a very high spatial resolution (0.84 and 0.98 m, respectively). The accuracy of the reconstructed datasets was assessed using a linear regression model with RMSE values of 3.55 and 3.44 m for the years 1967 and 1984, respectively. Our results show that SfM with MVS photogrammetry is effective in capturing complex and rugged topography from historical aerial photographs and at elevations ranging from the lowlands of the Delo Menna to the highest peak of the Bale Mountains of the Afro-alpine ecosystem of the Tulu Dimtu. Our datasets will help the scientific community address various research questions related to the Bale Mountains and afro-alpine ecosystems in general.

## Author contributions

MAM, BTH, GM, and TN created the research concept. MAM performed the photogrammetric processing and created the figures. MAM, LW and GM conducted the field work. MAM and DZ drafted the manuscript, and MAM, LW, DZ and BTH edited it. All authors contributed to the final version of the manuscript.

## Competing interests

The authors declare that there are no conflicts of interest.

## Acknowledgements

This research was funded by the German Research Council (DFG) in the framework of the joint Ethio-European Research Unit 2358 "The Mountain Exile Hypothesis: how humans benefited from and re-shaped African high-altitude ecosystems during Quaternary climatic changes".
We thank the Geospatial Information Institute of Ethiopia for providing us with the necessary data and documents; and Ethiopian Wildlife Conservation Authority, the Philipps University of Marburg, the Ethiopian Wolf

Project, and the Bale Mountains National Park for their cooperation and permission to conduct field work. We also very much appreciate the support of Mohammed Kedir, Hussein, Gash Kasim, Awol Assefa, Sofia, Wege Abebe, and Katinka Thielsen, without whom it would not have been possible to do the field work in the Bale Mountains. We thank Haileyesus Nega for his support in improving the figures. We thank Spaska Forteva for her help in arranging lab facilities. We also would like to acknowledge the anonymous reviewers.

**Financial support**

This work was supported by Deutsche Forschunggemeinschaft (DFG) Award no. NA 783/12-1, AOBJ 628803 through a project entitled "The mountain exile hypothesis: how humans benefited from and re-shaped African high-altitude ecosystems during Quaternary climatic changes" within the framework of Research Unit 2358.

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

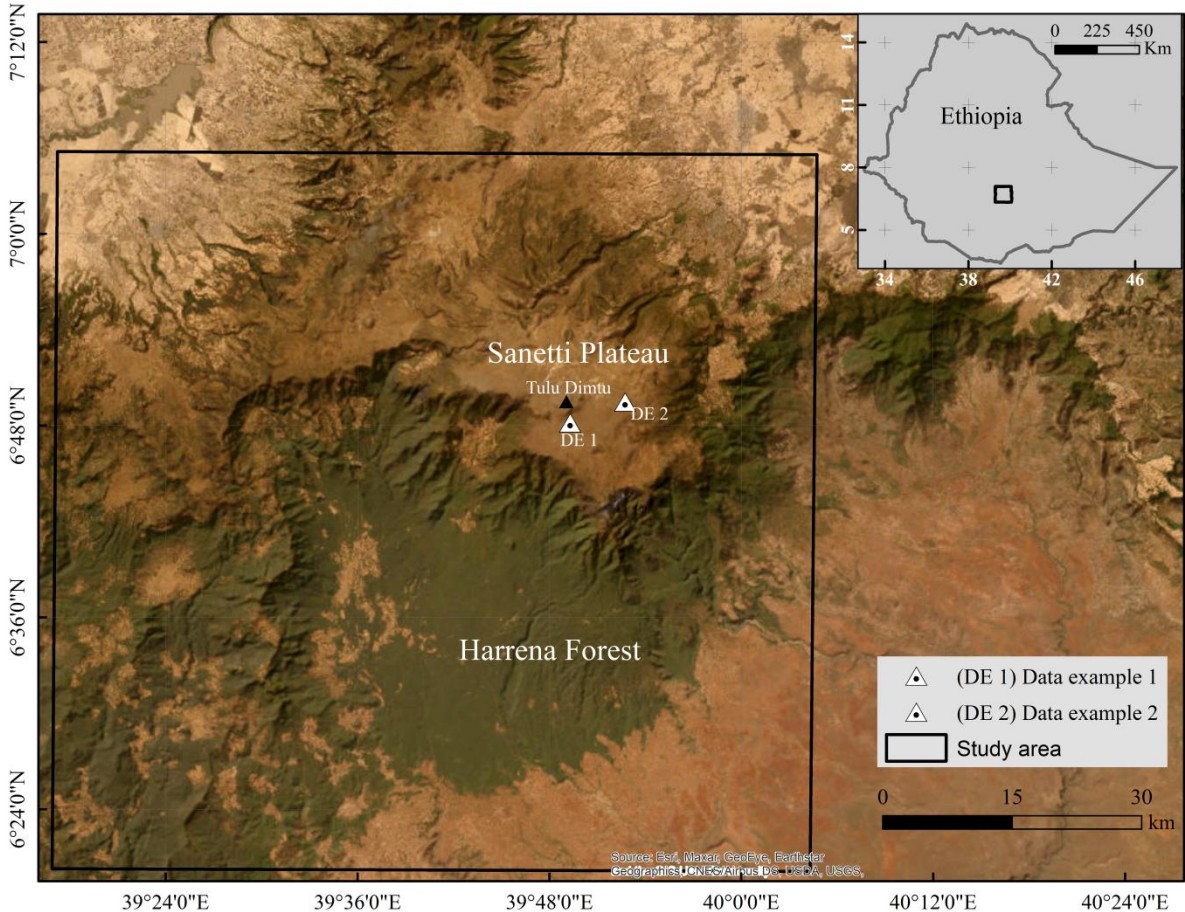

**Figure 1. Study area of the Bale Mountains in the southern Ethiopian Highlands, East Africa. (Source: Esri, Maxar, GeoEye, Earthstar Geographics, CNES/Airbus DS, USDA, USGS, AeroGRID, IGN, and the GIS User Community).**

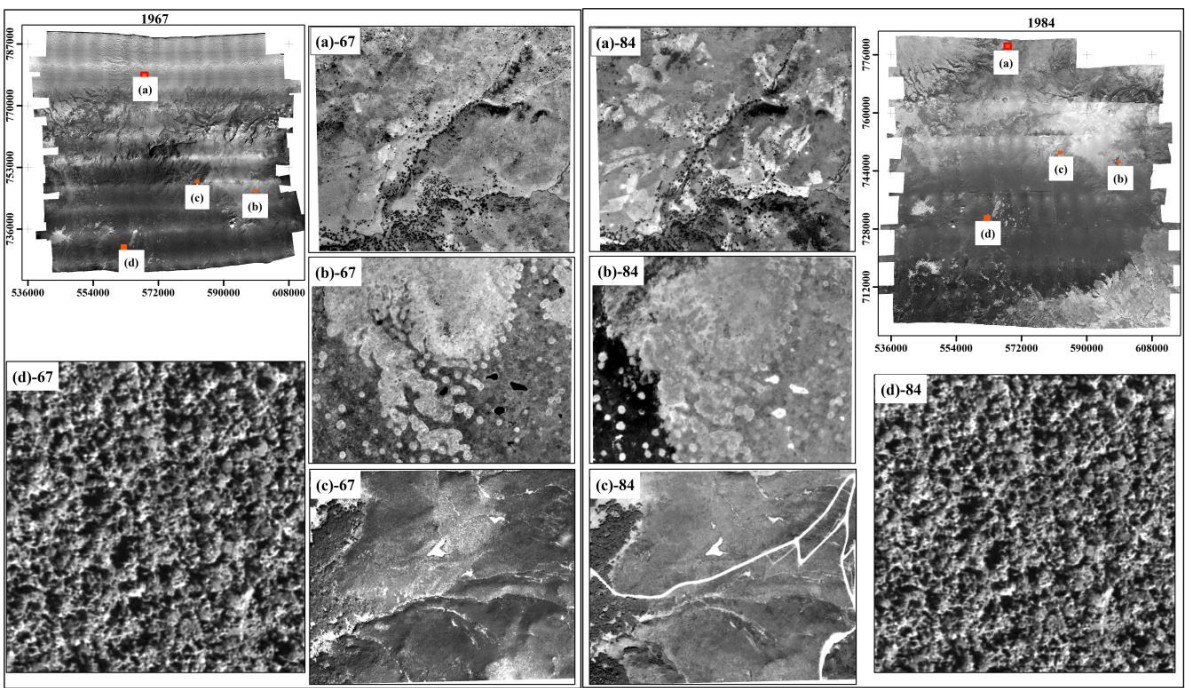

**Figure 2. Orthomosaics generated for 1967 (a)-67–(d)-67 and 1984 (a)-84–(d)-84 showing encroachment and cultivation of land in the north western portion of the study area, (a)-67 and (a)-84; forested area, (d)-67 and (d)-84; wetland area with earth mounds of ground-dwelling small mammals, (b)-67 and (b)-84; gravel road construction in 1984, (c)-84; and no visible road constructed in 1967, (c)-67.**

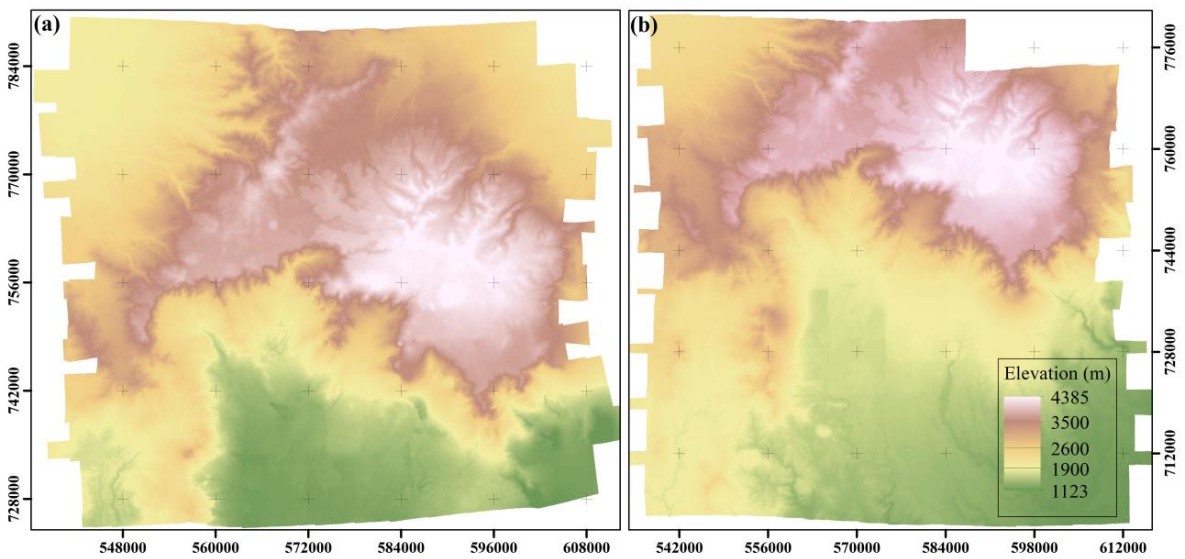

**Figure 3. Digital elevation models generated for the afro-alpine ecosystem of the Bale Mountains in Ethiopia based on scanned historical aerial photographs for the years 1967 (a) and 1984 (b).**

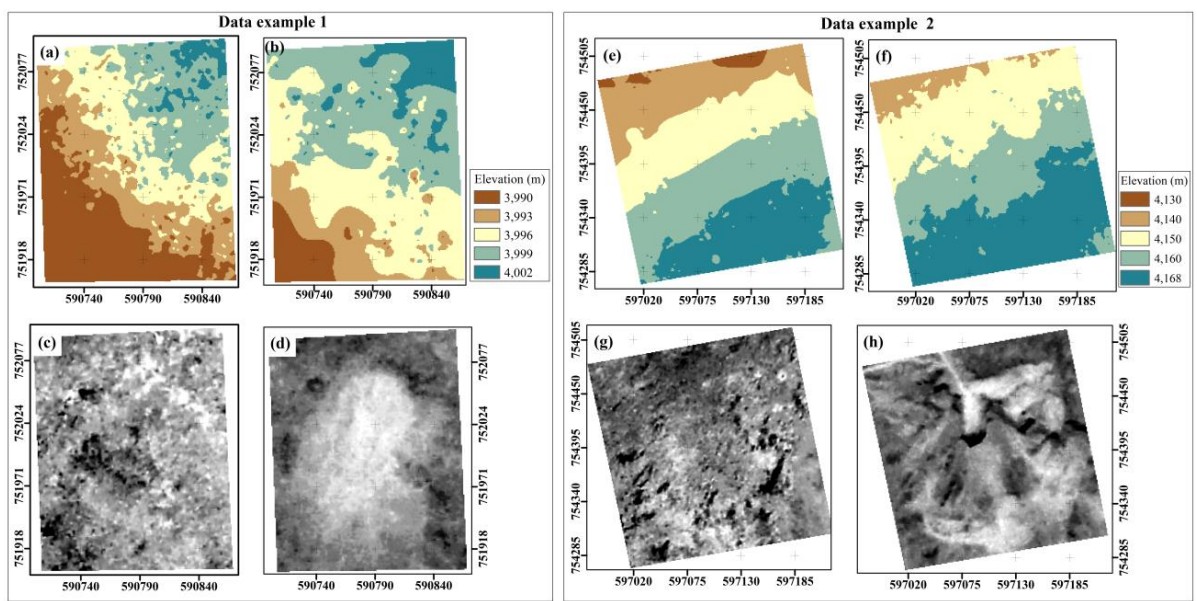

**Figure 4. Digital elevation model extent used for calculating the volumetric change in the study area for two example sites between the years 1967 and 1984: (a), (b), (e), and (f) for data examples 1 and 2 (see Fig. 1). Orthomosaics showing the planimetric extent of the extracted selected material: (c), (d), (g), and (h) at the centre of the study area, the Sanetti Plateau. The 1967 orthomosaics (c) and (g) depict no excavation. The 1984 orthomosaics (d) and (h) show the**

685 **excavation on the Sanneti Plateau. See also Fig. 1.**

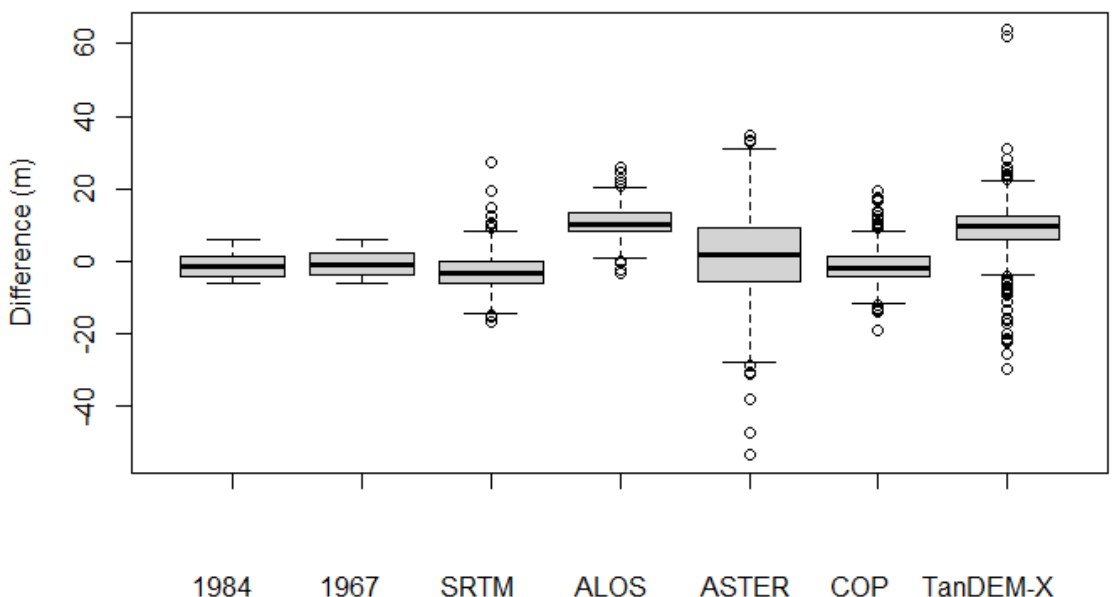

**Figure 5. Comparison of the distribution of error in elevation value differences between each GPS control point collected in the field and the DEMs generated for the years 1984 and 1967, and the readily available DEMs: ALOS PALSAR, SRTM, ASTER, Copernicus (COP), and TanDEM-X.**

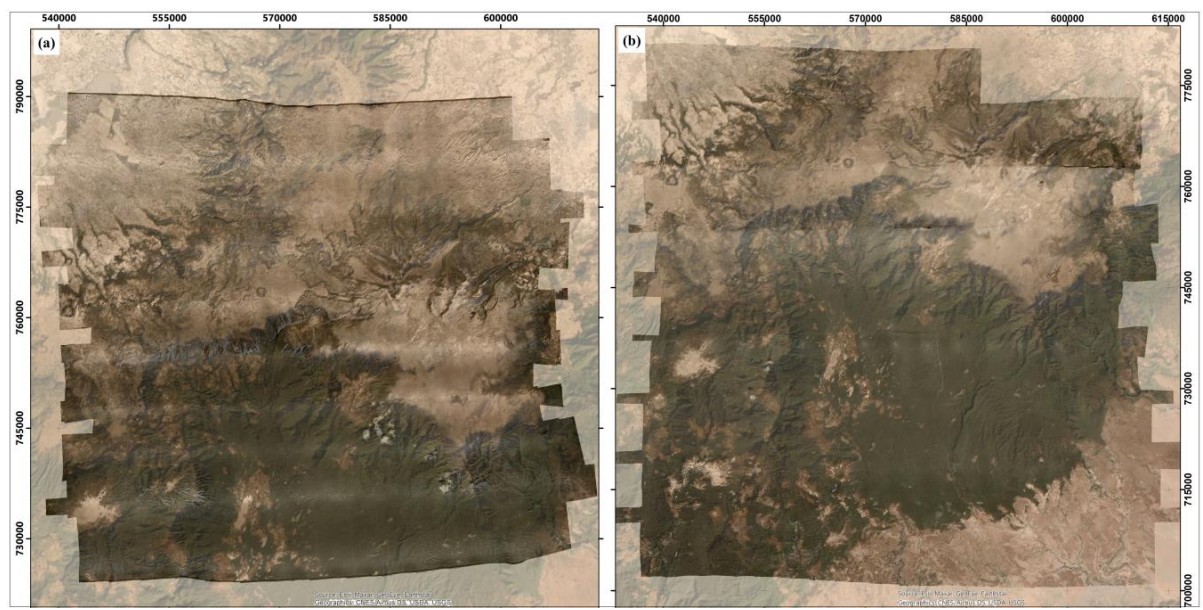

**Figure 6. Overlay snapshots of the orthomosaics for the year 1967 (a) and 1984 (b) over Global Google Earth images as a background (Background data source: Esri, Maxar, GeoEye, Earthstar Geographics, CNES/Airbus DS, USDA, USGS, AeroGRID, IGN, and the GIS User Community).**

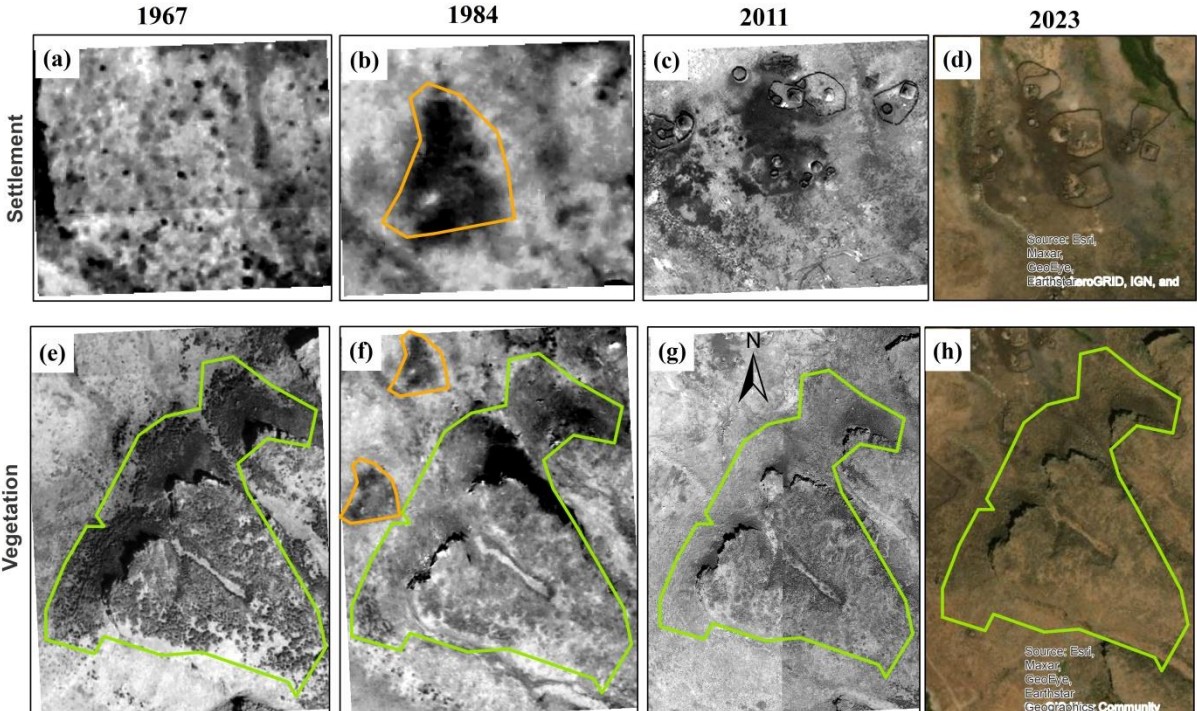

|  | 1967 | 1984 | 2011 | 2023 |
|--|------|------|------|------|
| Settlement | (a) | (b) | (c) | (d) |
| Vegetation | (e) | (f) | (g) | (h) |

**Figure 7. Settlers encroachment (top row) in the Bale Mountains National Park, Ethiopia (encircled in yellow colour) and associated vegetation deforestation (bottom row, encircled in green colour). Worldview-1 satellite image at 0.5 m resolution acquired in January, 2011, (Data sources 1967 and 1984: own data; Data source 2011: (Groos et al., 2021) and Data Source 2023: Esri, Maxar, GeoEye, Earthstar Geographics, CNES/Airbus DS, USDA, USGS, AeroGRID, IGN, and the GIS User Community).**

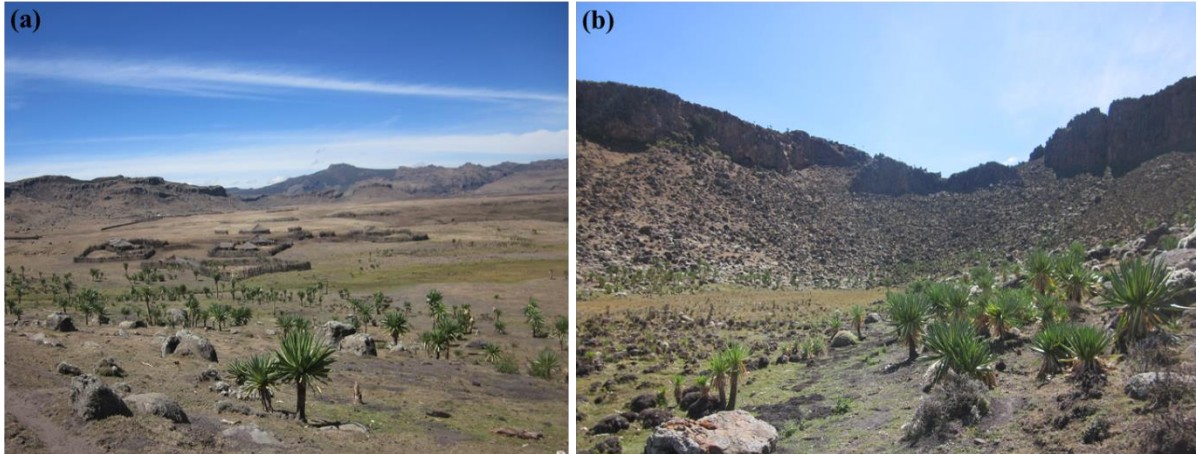

**Figure 8. Photograph taken in the field on January 24, 2020 showing (a) settlements and (b) *Erica* vegetation corresponding to the sites shown in figure 7. Location a: (6° 58' 42.54'' N, 39° 41' 30.24'' E), and location b: (6° 58' 37.18'' N, 39° 41' 41.48'' E).**

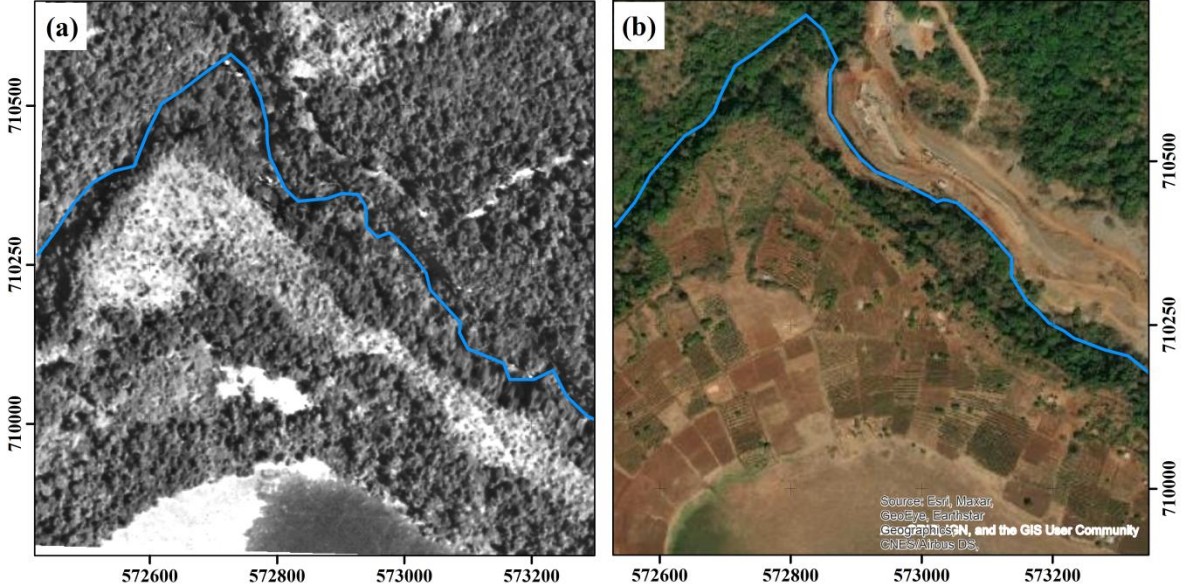

**Figure 9. The river course and riverside elevation have undergone significant changes since 1984, transforming from a forested river in that year (a) to their current appearance as seen in the Google Earth image (b). This specific location is situated in the southern part of the study area, approximately 21 km west of Delo Mena town. The imagery source includes contributions from Esri, Maxar, GeoEye, Earthstar Geographics, CNES/Airbus DS, USDA, USGS, AeroGRID, IGN, and the GIS User Community. The provided image was captured on December 15, 2020.**

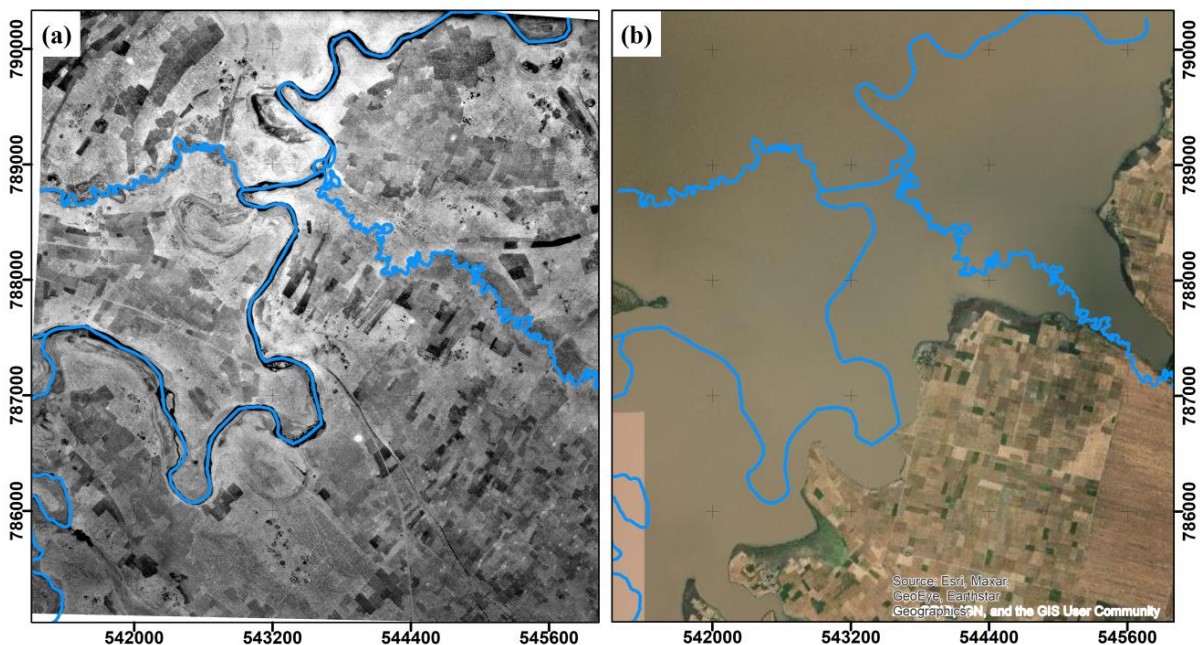

**Figure 10. A former river course (blue line) and agricultural area in 1967 (a) in the North-western corner of the study area changed to b) an artificial lake dam for hydroelectric power station currently viewed from global Google Earth image (Source: Esri, Maxar, GeoEye, Earthstar Geographics, CNES/Airbus DS, USDA, USGS, AeroGRID, IGN, and the GIS User Community) image taken on November 30, 2021.**

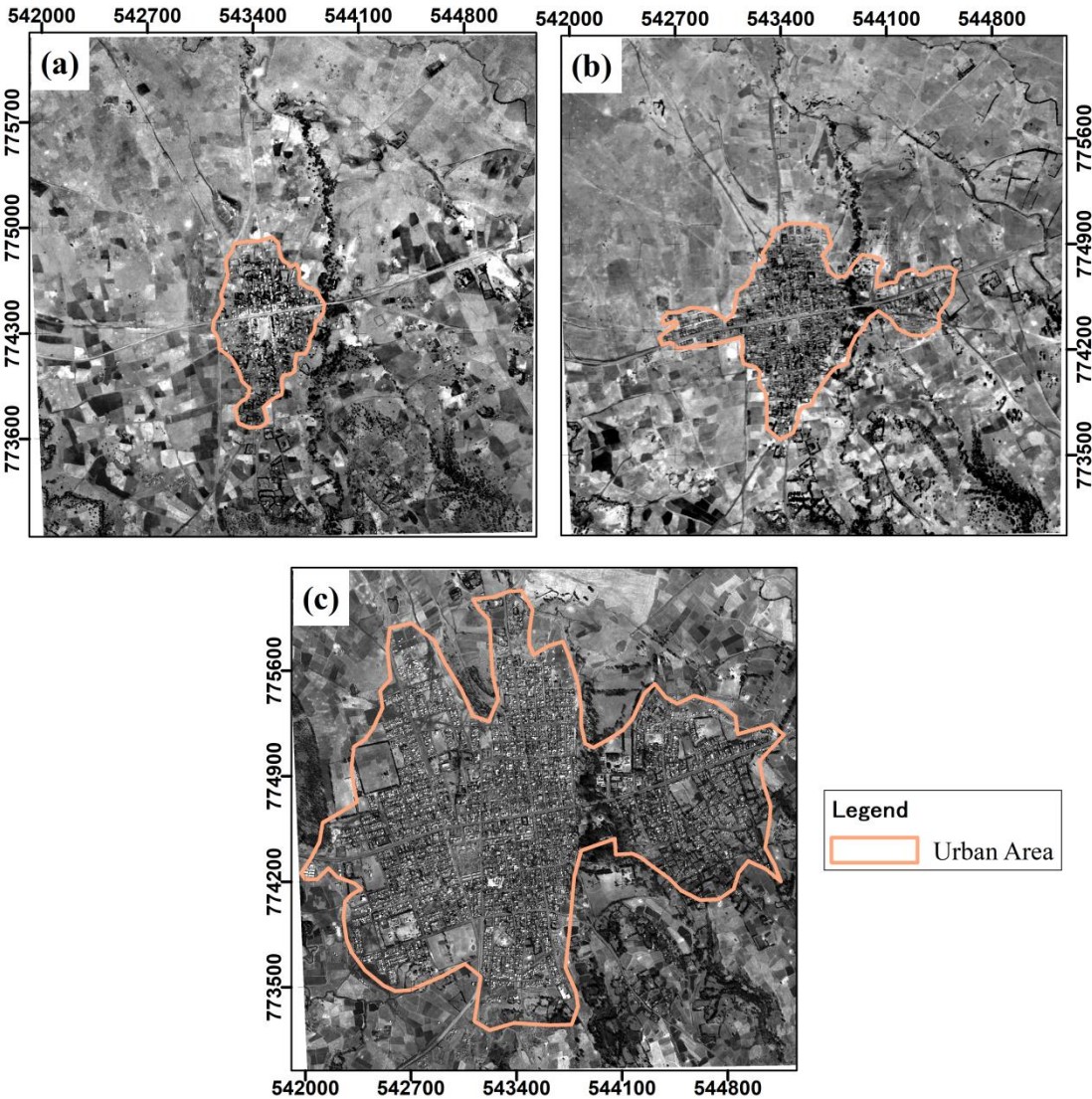

**Figure 11. Extent of the urban area of Adaba town in 1967 (a), 1984 (b) and 2017 (c) situated in the north-western extent of the study area. (Source: 1967 and 1984 here presented orthomosaic and for 2017 a SPOT-7 panchromatic image from Geospatial Information Institute of Ethiopia).**

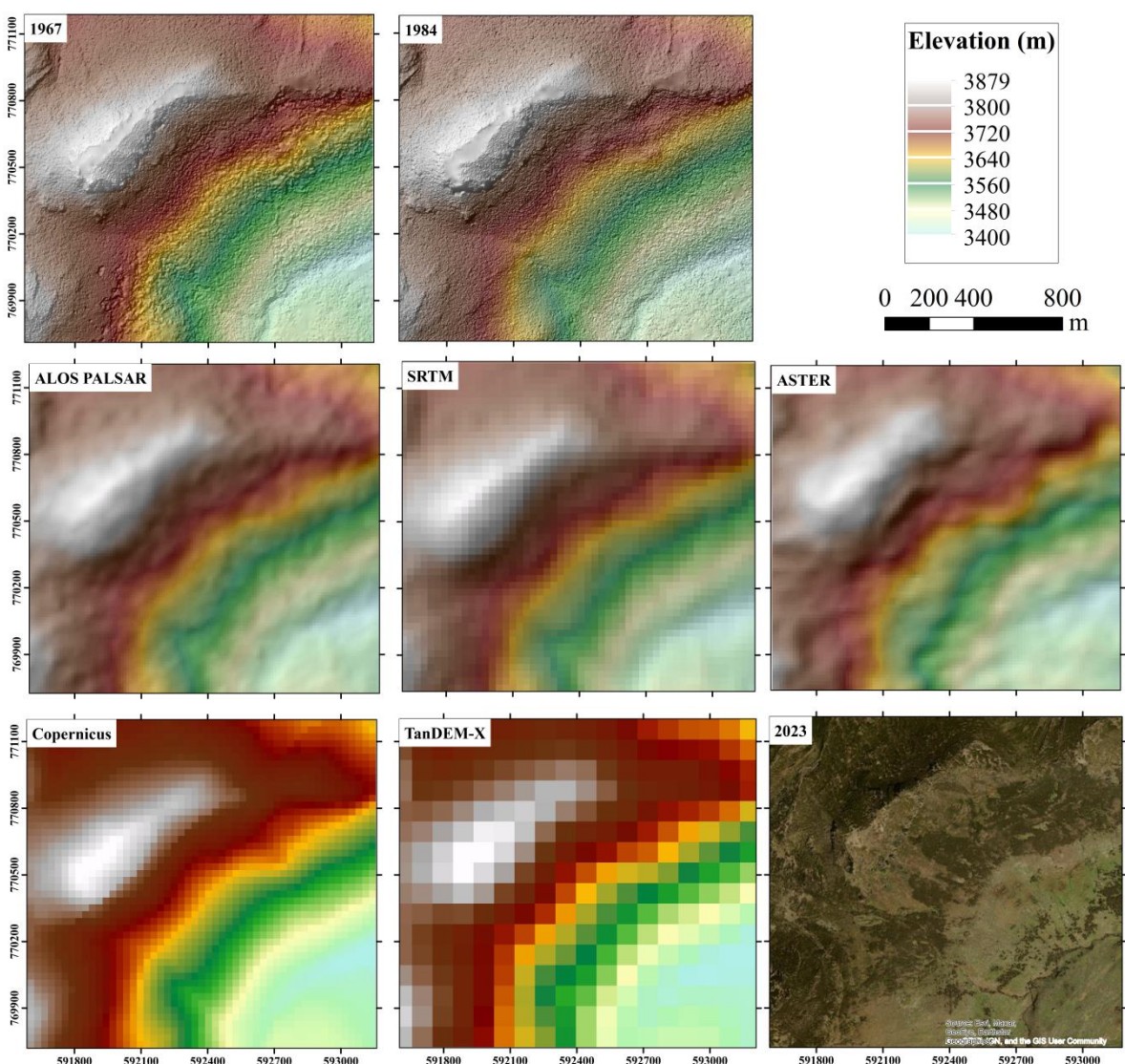

**Figure 12. Comparison of Digital Elevation Model (DEM) results in the northeastern study area, centered around Shaya Tika (6° 58' 16'' N, 39° 50' 15'' E) in the AJIDA region. The first row showcases DEMs from 1967 (0.84 m resolution) and 1984 (0.98 m resolution) left to right. The second row presents data from ALOS PALSAR (12.5 m resolution), SRTM (30 m resolution), and ASTER (15 m resolution) left to right. The third row features Copernicus DEM (30 m resolution), TanDEM-X DEM (90 m resolution), and the corresponding 2023 Google Earth image, left to**
**right. Google Earth image source: Esri, Maxar, GeoEye, Earthstar Geographics, CNES/Airbus DS, USDA, USGS, AeroGRID, IGN, and GIS User Community.**

770

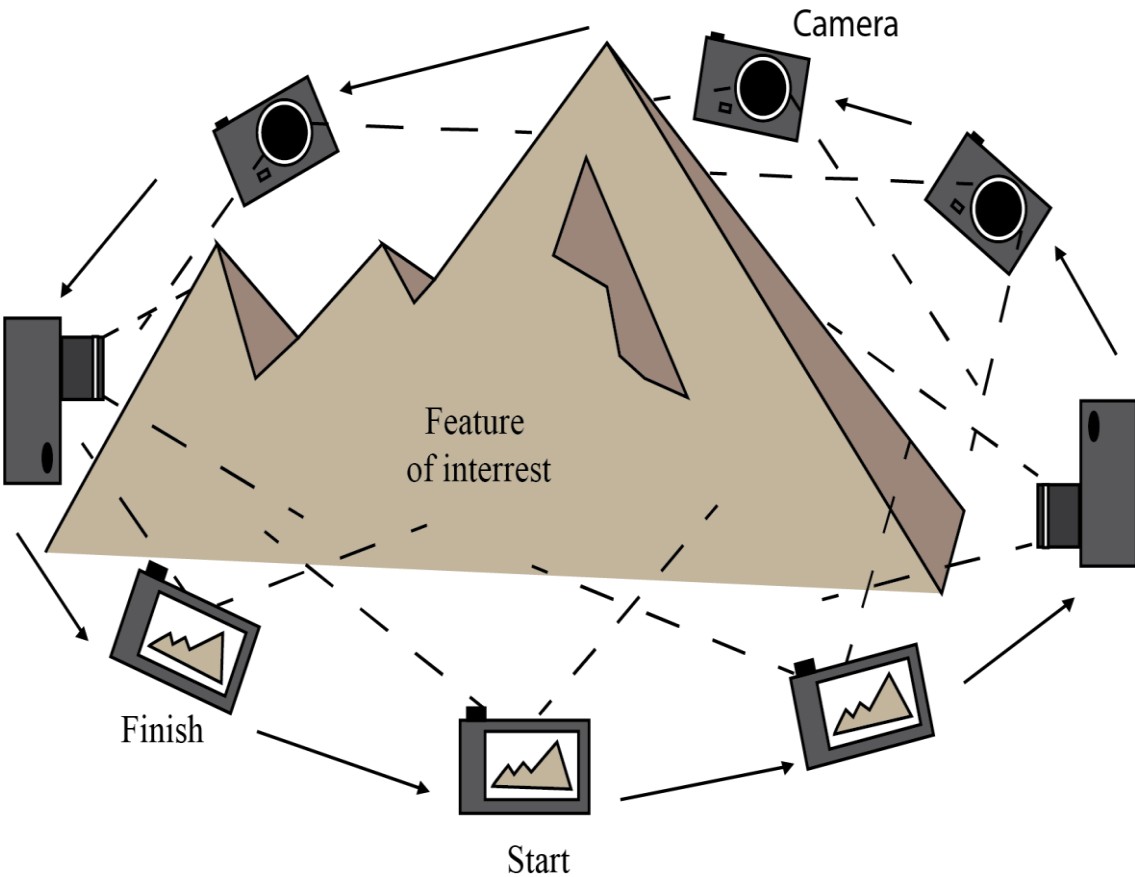

**Figure 13. Multi-view stereo model scheme: The process initiates with capturing multiple angles of the feature of interest, such as the topography of a landscape. Subsequently, these images are integrated using structure-from-motion techniques and algorithms through 3D software (e.g., Agisoft Metashape) on a computer, as outlined in methods in section 3, (Scheme by: Haileyesus Nega).**

780

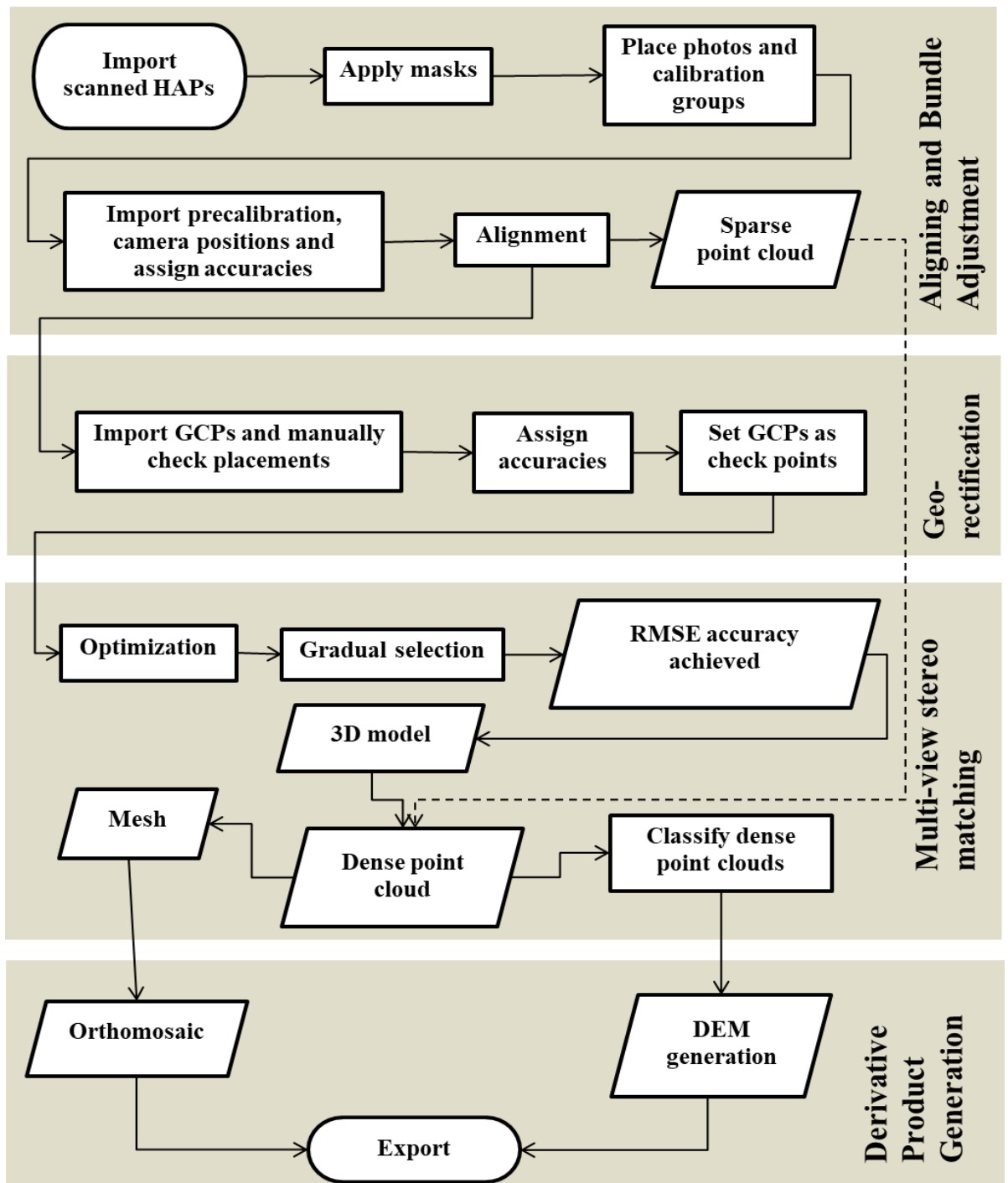

**Figure 14. Schematic workflow for Structure from Motion (SfM) Multi View Stereo (MVS) photogrammetry employed in the reconstruction of Digital Elevation Models (DEMs) and orthomosaics from historical aerial photographs (HAPs). Key terms: DEM - Digital Elevation Model, 3D - Three Dimensional, RMSE - Root Mean Square Error, GCPs - Ground Control Points. The workflow encompasses five principal stages: Alignment, Bundle Block Adjustment, Multiview Stereo Matching, Georectification, and Derivative Product Generation. Each stage involves a sequence of processes, inputs, and parameters crucial for software configuration. Alignment input parameters include: Accuracy (Highest), Generic Preselection (Yes), Reference Preselection (Source), Key Points Limit (100,000), Tie Point Limit (10,000), Exclude Stationary Tie Points (Yes). For Dense Point Cloud: Quality (Ultra High), Depth Filtering (Moderate), Reuse Depth Maps (Yes), Calculate Point Colours (Yes), Calculate Point Confidence (Yes). Mesh specifications comprise Source (Depth Maps), Quality (High), Face Count (Medium), Interpolation (Enabled), Reuse Depth Maps (Yes), Calculate Vertex Colors (Yes). Classify Dense Point Cloud: Max Angle (°) 37.5, Max Distance (m) 0.5, Cell Size (m) 30. For DEM: Source Data (Classified Dense Point Cloud), Interpolation (Disabled), Quality (Very High), Depth Filtering (Aggressive), Reuse Depth Maps (Yes). Orthomosaic settings encompass Resolution (m) 0 maximum, Surface (Mesh), Blending Mode (Mosaic), Hole Filling (Yes), Enable Culling (No), Refine Seamline (No).**

 **Table 1. Parameters, processing errors, and final product characteristics used and obtained in this study**

| Specification | 1967 | 1984 |
|---|---|---|
| Camera model | KC-1B | WILD RC 10 |
| Image size (pixels) | 12,000 x 12,000 | 12,000 x 12,000 |
| Focal length (mm) | default | 152.822 |
| Pixel size (μm) | 20 | 20 |
| Camera shutter type | Mechanical | Mechanical |
| Coverage (km$^2$) | 4,370 | 5,730 |
| Average flight height asl (m) | 9,448 | 7,600 |
| Number of images | 145 | 153 |
| Ground sampling distance (m/pix) | 0.84 | 0.98 |
| Number of tie points after filtration | 412,903 | 450,071 |
| Number of dense point clouds | 4,821,064,653 | 6,501,301,603 |
| Tie point RMS reprojection error (pix) | 1.49 | 0.68 |
| Average tie point multiplicity | 2.19 | 2.26 |
| Mean key point size | 4.90 | 4.75 |
| Dense cloud point density (point m$^{-2}$) | 1.41 | 1.03 |
| Number of control points | 49 | 27 |
| Total (3D) RMSE (cm) on control points | 4.24 | 0.88 |

**Table 2. External altitude (elevation) accuracy assessment showing statistical differences between control points and the respective points in the seven DEMs (Min = minimum elevation difference from field control points, 1Q = first quantile, Med = median, 3Q = third quantile; Max = maximum elevation difference from control points, RSE = Residual Standard Error, M R² = Multiple R-Squared, and RMSE = Root Mean Square Error)**

805

| Data | Min | 1Q | Med | 3Q | Max | RSE | M R$^2$ | RMSE |
|---|---|---|---|---|---|---|---|---|
| CP- 1984 | -5.56 | -2.79 | -0.04 | 2.41 | 7.40 | 3.22 | 0.9998 | 3.44 |
| CP- 1967 | -5.84 | -2.88 | -0.12 | 2.94 | 7.73 | 3.41 | 0.9998 | 3.55 |
| CP- ALOS PALSAR | -12.47 | -2.62 | -0.21 | 2.39 | 14.82 | 4.05 | 0.9997 | 11.38 |
| CP- SRTM | -14.03 | -2.92 | -0.49 | 2.71 | 29.95 | 4.87 | 0.9995 | 5.64 |
| CP- COPDEM | -16.54 | -2.73 | -0.39 | 2.51 | 21.49 | 4.63 | 0.9996 | 4.94 |
| CP-TanDEM-X | -36.82 | -2.84 | 0.63 | 3.79 | 53.74 | 7.88 | 0.9987 | 11.9 |
| CP- ASTER | -54.58 | -7.76 | -0.05 | 7.25 | 32.40 | 11.42 | 0.9974 | 11.54 |

810