# Peer review of "High-resolution digital elevation models and orthomosaics generated from historical aerial photographs (since the 1960s) of the Bale Mountains in Ethiopia"

_Earth System Science Data, 2022_

## Author Comment (AC1)

**Title: High-resolution digital elevation models and orthomosaics generated from historical aerial photographs (since the 1960s) of the Bale Mountains in Ethiopia**

We would like to thank the first Referee for his/her time in reviewing our manuscript and for providing valuable suggestions. Our response to your comments are colored in blue, the original comments are colored in black. During the revision process of this article we received additional support and advice from "Luise Wraase". Therefore, in the revised manuscript we have decided to include her as a co-author of the paper. We hope that our responses qualify us to submit a revised version of the manuscript.

**Referee: 1**

Comment: The motivation of the study presented in the introduction mainly focuses on thematic aspects, such as ecosystem, biodiversity, land use and climate change research in Ethiopia and the need for historic high-resolution spatial data (DEM and surface cover) for change and impact analysis. However, a profound assessment of globally available high-resolution satellite remote sensing data is missing. Such data include the KH-1 to KH-9 Corona (1960-1986), IRS (since 1995) and Digital Globe (since 1999) programs among others. The authors need to embed and compare their two snapshots in time (in) to a bigger time line and thus, demonstrate that their data spatially align to the globally available datasets in order to ensure data continuity in space and time which is listed as the first priority in using remote sensing data for biodiversity and related purposes (Tuner et al., 2015).

> Response: We added one paragraph (lines 65-75) on globally available high-resolution satellite remote sensing data and the spatial and temporal alignment of our data compared to the ones with global coverage. However, the Corona declassified photographs have in general/worldwide limitations like invisibility of film reference data, low texture and film saturation, inconsistent overlap; and scanning artefacts (Ghuffar et al., 2022). In contrast, our data is fully processed and ready to use. Satellite imageries of very high resolution like the IRS and Digital Globe are commercial, therefore they are not freely accessible, in contrast to our data.

Comment: Moreover, the paper is missing a relevant case study demonstrating the value of the datasets for one of the significant purposes mentioned in the introduction. Just picking one case – volume difference at one section of a gravel road – is not sufficient for that. Such a case study also needs to include the comprehensible demonstration of alignment with other globally available high-resolution data sets.

> Response: We added four additional case studies as examples further demonstrating the value of our dataset: settlement expansion, river and river side altimeter change, and surface geomorphology and urban expansion extent (lines 293-314). We also added a demonstration of the alignment of the generated datasets to currently available global high resolution images (Fig. 6).

Comment: Overall, the paper is written rather 'minimalistic' whereas most of the content is dedicated to standard technical descriptions (sections 2 and 3). The lack of more in-depth evaluation of the data is also reflected in the content and the quality of the figures as well as the results and discussions parts of the paper. From the description is unclear which data are used for accuracy assessment. The listed DEM data sets cannot be used for accuracy assessment since they are of much coarser spatial resolution. Later in the paper, the authors state the use of independent GCP's for accuracy assessment. If this is the case, this information needs to be mentioned already in this section. The authors also need to consider using TanDEM-X DEM information (12.5 m resolution) in their study.

> Response: We improved the quality of figures (Figs. 1-5) by adjusting the alignment of text and figure labels indentation, adjusting the latitude and longitude of grids, increasing contrast and brightness, and increasing export resolution. To enhance the overall level of detail we added nine more figures (Fig. 6-14) and expanded the result and discussion parts (lines 265-356).
> We furthermore reworked the section of the accuracy assessment (lines 316-356, Fig. 5). The independent field-collected GPS points (509 points) were used for external accuracy assessment (lines 316-318) and we updated the section that among the GCPs, 12 out of the 49 GCPs for 1967 and 7 out of the 27 GCPs for 1984 were used for internal accuracy assessment (lines 334-335). Freely available DEMs (SRTM 30 m, ASTER 15 m, and ALOS PALSAR 12.5 m) were used for additional comparison. We also added the Copernicus DEM and TanDEM-X DEM (30 m and 90 m resolution, respectively) as new and latest datasets for comparison in addition to the above mentioned three readily available DEMs.
>
> Since the TanDEM-X 12.5 m resolution dataset is only commercially available, we did not include it in the accuracy assessment. However, the TanDEM-X 90 m is openly available and we used it for the assessment in addition to Copernicus DEM (30 m) based on the TanDEM-X mission. However the spatial resolution of ALOS PALSAR DEM is the same as the TanDEM-X DEM.

Comment: Ground control points: Have these GCP's only be used for reference purposes or has a subset of them also be used for independent accuracy assessment of the resulting DEM's? Such an approach using control information of significantly higher accuracy would be needed for a state-of-the-art accuracy assessment.

Response: 509 GCPs collected from the field were used for external independent accuracy assessment of the resulting DEMs (lines 316-318). Visually selected ground control points from the images were used for processing the scanned historical aerial photographs and a subset from them for internal accuracy assessment (now better described in lines 135 -137 and 334 - 335). We clarified the difference between them in section 2 (lines 135-139).

Comment: Results and discussion: Do the achieved accuracies represent internal accuracies or do they relate to external reference systems too? At this point it becomes evident that the methodological section 3 of the paper lacks a subsection describing the approach used for accuracy assessment in this study which would need to include the internal and the external accuracies as well as the alignment with other high-resolution data sets as already mentioned above. So far, there is only a very brief methodological description at beginning of section 4.2. The thematic discussion in section 4.1 is very minimal and needs to be extended (see comment under general remarks).

Response: The RMSE accuracy values of 3.55 and 3.44 for 1967 and 1984, respectively, relate to external reference systems (line 333). However, the accuracy has additionally been assessed internally.

For internal accuracy assessment, among the 76 on-images selected GCPs, 19 of them were selected and used as control points in Agisoft metashape professional software. The achieved accuracies represent internal accuracy (we added more details in lines 334-338 and Table 1).

We extended the description of the external accuracy assessment in lines 316–333 and Table 2.

Moreover, we extended section 4.1 by e.g presenting four more use case studies (lines 293-314).

We also extended section 4.2 on quality assessment (lines 316-356) to better distinguish between internal and external assessments.

Comment: The quality assessment part proofs the high quality of the positional accuracy of the derived data sets. However, analysis of height accuracy seems to be missing and should be easy to perform using the available GCP's.

Response: Thank you for highlighting the positional accuracy quality of our data set. The height accuracy can be found in lines 328-329, and Tables 1 & 2.

Comment: The further analysis of the quality of readily available DEM's based on the same GCP's is interesting but not an accuracy assessment related to the data sets derived in this study. However, these data sets could partly be used in order to analyze the alignment with external data.

Response: As mentioned above, we first assessed the quality of our generated datasets using external GPS points sampled in the field as GCPs. Additionally, we think that our comparison to readily available DEMs is interesting and further demonstrates the high quality of our dataset (lines 324-326).

Comment: The current discussion between the different external DEM's lacks clarity; the authors need to describe methodologically sound which parameters have been used in order to assess the accuracy of their data sets and the different external DEM's. Moreover, RMS-errors of around 10 meters are not huge for DEM's derived from medium resolution satellite data.

Response: To improve the clarity of the discussion section, we added more sentences describing the use of parameters and accuracy assessment (lines 320-324) and adjusted the text about the accuracy of the external DEMs (lines 339-356, 351).

---

## Author Comment (AC2)

**Title: High-resolution digital elevation models and orthomosaics generated from historical aerial photographs (since the 1960s) of the Bale Mountains in Ethiopia**

We would like to thank the second Referee for his/her time in reviewing our manuscript and for providing valuable suggestions. Our response to your comments are colored in blue, the original comments are colored in black. During the revision process of this article we received additional support and advice from "Luise Wraase". Therefore, in the revised manuscript we have decided to include her as a co-author of the paper. We hope that our responses qualify us to submit a revised version of the manuscript.

**Referee: 2**

Comment: This article presents an intriguing approach to constructing a digital elevation model of the Bale Mountains in Ethiopia using high-resolution historical images. However, for the manuscript to be considered for acceptance, there is significant room for improvement in various aspects, including overall structure, abstract, introduction, materials, results, and discussion. I offer the following suggestions in hopes of providing constructive feedback.

I am concerned about the paper's originality, as a related article has already been published: Nyssen et al., 2022, Online Digital Archive of Aerial Photographs (1935–1941) of Ethiopia. I would appreciate clarification on the innovative aspects of this manuscript compared to the paper. Please provide a detailed explanation of the data and reconstruction algorithm employed for the digital elevation model.

> Response: Our paper is quite different from Nyssen et al. 2022 for several reasons:
> 1) Methodologically we applied Structure from Motion Multiview Stereo Photogrammetry using the software Agisoft metashape professional. This allowed us to calibrate the images using information like calibrated focal length, principal points, fiducial marks, flying altitude, ground control points and camera exposure stations. However, Nyssen et al. 2022 used the software Photoscan, which is a former version of Agisoft metashape professional and did not include the above mentioned calibration information as an input.

2) We reconstructed DEMs and Orthomosaics for two points in time (1967 and 1984). Thus, there is no temporal overlap with the data from Nyssen et al. 2022, who provide data from 1935-1941.

3) We provide data for the Bale Mountains i.e. the second highest place of Ethiopia. The altitudinal range of our study area ranges from 977 to 4,377 m asl, and hence, there is no spatial overlap with Nyssen et al. 2022.

4) The spatial coverage of our study area is approximately 5,370 km$^2$ (line 100). However, the spatial extent of Nyssen et al. 2022 is smaller compared to ours i.e. 58 km2.

5) We did the georectification fully in Agisoft metashape professional software, in which both DEMs and orthomosaics were reconsructed. However, Nyssen et al. 2022 did the georectification in ArcMap and generated solely the orthomosaic.

6) The RMSE accuracy achieved for the reconstructed DEMs of 1967 and 1984 is 3.55 and 3.44 m, respectively. However, the RMSE value of Nyssen et al. 2022, is about 30 m.

We added a figure illustrating our workflow (Fig. 14) and wrote an additional paragraph describing the methods in more detail (lines 205-260).

Comment: The abstract is generally well-written, but it should emphasize the innovative aspect of this study, specifically the reconstruction of a detailed digital elevation model, as well as the method utilized. Since this paper offers a public dataset, please include validation methods and accuracy measures.

Response: We edited and restructured the abstract by incorporating the methodology for reconstructing our detailed digital elevation model, the validation methods and the accuracy measures (lines 27-32).

Comment: The introduction section could be further strengthened by emphasizing the importance of a detailed digital elevation model and addressing the limitations of previous research. It is worth mentioning that three-dimensional modeling has been a relatively mature technology dating back to the 1960s. What methods were employed before then to construct these models? Additionally, provide a brief introduction to how the technology used in this study handles image data. While the structure of this section is clear, further clarification is needed regarding the importance and significance of the research area and research questions.

Response: We strengthened the introductory part by adding more paragraphs giving the historical background, details about previous research, emphasizing the relevance of DEMs, and by adding more information about the applied methodology and significance of the study region (lines 65-88).

Comment: The second and third sections can be combined into one, and I highly recommend using a flowchart to clearly explain the entire reconstruction process, including data preprocessing, algorithm modules, and so on.

Response: We kept the second (Material and methods, now seven subsections) and third section (Data processing, now extended by more details on the methods in seven paragraphs) separately because we believe that this enhances the clarity of our manuscript. We added a flowchart to better explain the reconstruction process (Fig. 14).

Comment: The materials and methods section needs substantial revision. Although the paper mentions numerous data preprocessing methods and digital elevation model reconstruction algorithms, many aspects require further elaboration. For instance, consider providing more detail on calibrated focal length, principal points, fiducial marks, and ground control points, accompanied by corresponding images.

Response: Done. We now give more details about calibrated focal length, principal points, fiducial marks, and ground control points (lines 166-170, line 179-181, line 186-189 and line 192-194). Additional images and tables describing more details can be found in the supplementary material (Supplementary Fig. S1-S4 and table S1-S7).

Comment: The Structure-from-Motion Multi View Stereo Photogrammetry algorithm mentioned in the paper should be described in detail, including its processing specifics and core formulas. Highlight the method's specific advantages in your research, such as efficiency, accuracy, and other aspects of the reconstruction process. Overall, these two sections lack detail, and providing more information can significantly enhance the paper's readability.

Response: The Structure-from-Motion Multi View Stereo Photogrammetry is now described in detail in a new paragraph and texts (lines 205-260) including processing specifics and the core formula the algorithm uses to reconstruct the DEMs and orthomosaics. In addition, the overall process is now included as a flowchart (Fig. 14).

Comment: The results and discussion section also need substantial revision. The paper does not highlight many points of interest, such as the study area's topography, vegetation, etc. Consider providing Google Earth images of the study area. From a validation standpoint, the dataset appears to have produced good results.

Response: Thank you for highlighting the good quality of our dataset. We substantially revised the results and discussion section. We added more text regarding the topography and vegetation of the study area (lines 118-130 and 265-271). We added a figure that shows the orthomosaics overlayed on Google

Earth images (Figs. 1 and 6). Additionally, we extended the possible and recent use cases in section 4.1 (lines 293-314).

Comment: I am curious about the specific validation method, as the elevation datasets being compared differ in time and spatial resolution. When performing comparative validation, zoom in on local maps to emphasize the differences between various DEM datasets. Additionally, discuss the influence of vegetation and terrain on mapping accuracy during the elevation model creation process. This section needs significant expansion; otherwise, it will be difficult to attract attention.

Response: We collected georectification and validation points where no landscape change occurred. In order to minimize the effects of vegetation during reconstruction of the elevation model, we used parameters of maximum angle, maximum distance and cell size values to classify the generated dense point clouds to ground points. We added a zoomed-in figure to show the difference between freely available DEMs and the reconstructed DEMs (Fig. 12, more descriptions in lines 320-326). However, the terrain and vegetation cover can affect the accuracy of the generated DEM, which is discussed in lines 275-277.

Comment: Lastly, please rewrite the conclusion section.

Response: Done (lines 388-399).

Comment: Other suggestions include:

Adjust the font size and style of all figures and tables for easier reading.

Response: The font size and style of all figures and tables was adjusted, according to the journal's standard template of Copernicus ESSD.

Comment: Figure 2 may be challenging for readers to interpret due to its grayscale appearance. Please provide appropriate satellite images.

Response: The appearance and visibility of Fig. 2 was improved by increasing brightness and contrast, and adjusting the labels.

Comment: Correct the latitude and longitude information corresponding to Figure 3 and Figure 4.

Response: Done.

Comment: Improve Figure 5's visual appeal.

Response: Done. The figure was improved by increasing length, width and resolution values.

Comment: Unify the paper's font style, such as the red font at line 222.

Response: Done.